# New Multifunctional Agents for Potential Alzheimer’s Disease Treatment Based on Tacrine Conjugates with 2-Arylhydrazinylidene-1,3-Diketones

**DOI:** 10.3390/biom12111551

**Published:** 2022-10-24

**Authors:** Natalia A. Elkina, Maria V. Grishchenko, Evgeny V. Shchegolkov, Galina F. Makhaeva, Nadezhda V. Kovaleva, Elena V. Rudakova, Natalia P. Boltneva, Sofya V. Lushchekina, Tatiana Y. Astakhova, Eugene V. Radchenko, Vladimir A. Palyulin, Ekaterina F. Zhilina, Anastasiya N. Perminova, Luka S. Lapshin, Yanina V. Burgart, Victor I. Saloutin, Rudy J. Richardson

**Affiliations:** 1Postovsky Institute of Organic Synthesis, Urals Branch of Russian Academy of Sciences, Yekaterinburg 620990, Russia; 2Institute of Physiologically Active Compounds at Federal Research Center of Problems of Chemical Physics and Medicinal Chemistry, Russian Academy of Sciences, Chernogolovka 142432, Russia; 3Emanuel Institute of Biochemical Physics Russian Academy of Sciences, Moscow 119334, Russia; 4Department of Chemistry, Lomonosov Moscow State University, Moscow 119991, Russia; 5Department of Environmental Health Sciences, University of Michigan, Ann Arbor, MI 48109, USA; 6Department of Neurology, University of Michigan, Ann Arbor, MI 48109, USA; 7Center of Computational Medicine and Bioinformatics, University of Michigan, Ann Arbor, MI 48109, USA; 8Michigan Institute for Computational Discovery and Engineering, University of Michigan, Ann Arbor, MI 48109, USA

**Keywords:** Alzheimer’s disease, acetylcholinesterase, butyrylcholinesterase, inhibitors, 2-tolylhydrazinylidene-1,3-diketones, tacrine conjugates, propidium displacement, antioxidant activity, biometals, ADMET prediction

## Abstract

Alzheimer’s disease (AD) is considered a modern epidemic because of its increasing prevalence worldwide and serious medico-social consequences, including the economic burden of treatment and patient care. The development of new effective therapeutic agents for AD is one of the most urgent and challenging tasks. To address this need, we used an aminoalkylene linker to combine the well-known anticholinesterase drug tacrine with antioxidant 2-tolylhydrazinylidene-1,3-diketones to create 3 groups of hybrid compounds as new multifunctional agents with the potential for AD treatment. Lead compounds of the new conjugates effectively inhibited acetylcholinesterase (AChE, IC_50_ 0.24–0.34 µM) and butyrylcholinesterase (BChE, IC_50_ 0.036–0.0745 µM), with weak inhibition of off-target carboxylesterase. Anti-AChE activity increased with elongation of the alkylene spacer, in agreement with molecular docking, which showed compounds binding to both the catalytic active site and peripheral anionic site (PAS) of AChE, consistent with mixed type reversible inhibition. PAS binding along with effective propidium displacement suggest the potential of the hybrids to block AChE-induced β-amyloid aggregation, a disease-modifying effect. All of the conjugates demonstrated metal chelating ability for Cu^2+^, Fe^2+^, and Zn^2+^, as well as high antiradical activity in the ABTS test. Non-fluorinated hybrid compounds **6** and **7** also showed Fe^3+^ reducing activity in the FRAP test. Predicted ADMET and physicochemical properties of conjugates indicated good CNS bioavailability and safety parameters acceptable for potential lead compounds at the early stages of anti-AD drug development.

## 1. Introduction

Alzheimer’s disease (AD) is the most common form of dementia in old age. It is currently considered a modern epidemic because of its increasing prevalence worldwide and serious medico-social consequences, including the economic burden of treatment and patient care. This situation is due to a multiplicity of interacting factors including the steadily growing population, increase in life expectancy, duration of the course of the disease, severe disability of patients requiring their hospitalization or expensive in-home care, and the lack of effective therapy [1,2]. Therefore, the development of new effective multifunctional agents that can act simultaneously on several targets thought to be involved in AD pathogenesis is one of the most urgent and important tasks of medicinal chemistry and pharmacology [3].

One of the main characteristics of AD is cholinergic deficiency. A loss of cholinergic innervation in the cerebral cortex of patients with this disorder is an early pathogenic event correlated with cognitive impairment. The severity of dementia has been found to have a positive correlation with the extent of cholinergic loss. This evidence led to the formulation of the “cholinergic hypothesis” and the development of cholinesterase inhibitor therapies for symptomatic improvement in AD patients [4].

Initially, cholinergic therapy for AD was directed toward inhibition of acetylcholinesterase (AChE, EC 3.1.1.7) as the main enzyme that hydrolyzes the neurotransmitter acetylcholine [5]. The therapeutic effect of AChE inhibitors arises from the increased concentrations and duration of action of acetylcholine in cholinergic synapses [6,7]. Butyrylcholinesterase (BChE, EC 3.1.1.8) is also involved in the hydrolysis of acetylcholine and can compensate for some of the functions of AChE, thereby optimizing cholinergic neurotransmission [8,9,10,11,12]. As the disease progresses, the activity of BChE gradually increases while the activity of AChE decreases [12,13]. Consequently, BChE has gained importance as a therapeutic target for reducing cholinergic deficiency [14,15,16].

Currently, there are no treatments that effectively mitigate the underlying pathogenic mechanisms for AD. Modern therapeutic strategies for AD consist of three cholinesterase inhibitors: donepezil, rivastigmine and galantamine, and the NMDA receptor antagonist memantine. Unfortunately, the effects of these agents are merely palliative. They partially compensate for declining cognitive function but they are not able to stop the development of the neurodegenerative process [7]. Such a small choice of drugs for AD therapy is due to the multifactorial nature of this disease, a realization that has fostered interest in identifying contributing mechanisms and discovering agents capable of attenuating these processes simultaneously [17,18,19].

One of the best-characterized pathogenic processes of AD is β-amyloid (Aβ) protein aggregation and deposition in the brain [20,21]. In addition to the classical function of acetylcholine hydrolysis, AChE has proaggregant properties toward Aβ via participation of the peripheral anionic site (PAS), which interacts with soluble Aβ peptides promoting their aggregation [22,23]. Thus, compounds interacting with the AChE PAS are potential antiaggregating agents [24,25]. Moreover, BChE is also thought to be involved in the formation and/or maturation of Aβ plaques, thereby contributing to AD pathogenesis [26,27,28,29,30,31]. Therefore, inhibitors of BChE and inhibitors of both cholinesterases are of particular interest from the standpoint of a dual strategy: increasing the concentration of acetylcholine and ameliorating β-amyloid aggregation.

Another important mechanism in many neurodegenerative diseases is oxidative stress, characterized by an imbalance between the formation of reactive oxygen or nitrogen species and their inactivation by antioxidant systems [32,33,34]. The brain is the most sensitive organ in the body to the damaging effects of free radicals, and this vulnerability increases with age [32,35]. The association between AD and oxidative stress is widely investigated as a potential therapeutic target [36,37,38,39,40], and the design of cholinesterase inhibitors with antioxidant properties is considered a promising direction in the development of multifunctional drugs for AD treatment [41,42,43,44,45].

One of the plausible mechanisms contributing to AD pathogenesis is imbalance of brain homeostasis of certain ions, e.g., Cu^2+^, Zn^2+^, and Fe^2+^. The content of these brain biometals increases 3- to 7-fold during the progression of AD [46]. The accumulation of metals contributes to their binding to Aβ, leading to its increased aggregation [47,48]. Moreover, Cu^2+^ and Fe^2+^ support the production of reactive oxygen species and increase of oxidative stress, thus promoting neurotoxicity [49,50]. Therefore, selectively reducing brain concentrations of metals with chelating agents is one of the therapeutic approaches proposed for the treatment of AD [51,52].

Considering the multifactorial nature of AD, a promising therapeutic approach is the development of multitarget drugs having a complex effect on several biological targets responsible for the pathogenesis of this disease [53,54,55,56,57]. One of the design strategies for multitarget drugs is to use a molecular spacer to link together two pharmacophores that are active against two or more different biological targets, and a well-known anticholinesterase drug molecule is often used as one of the pharmacophores [58,59,60,61,62].

Among the anticholinesterases, tacrine was approved in 1993 as the first drug for the treatment of AD [63,64]. However, its serious side effects, such as hepatotoxicity, led to its withdrawal from the market. To improve its activity and reduce its toxicity, new derivatives of tacrine have been designed and synthesized [65,66].

Many studies have been aimed at modifying tacrine by creating hybrid conjugate compounds linked through a spacer with various pharmacophores that promote the interaction of the conjugate inhibitor with both the catalytic active site (CAS) of AChE and its PAS, thus blocking AChE-induced Aβ aggregation [41,67,68]. In addition, various heterocyclic compounds, e.g., hydroxyquinolines [69] and coumarins [70,71], as well as open-chain fragments such as amino acid derivatives [72,73] and curcumin [74] have been used as a second pharmacophore. These tacrine conjugates combined AChE and BChE inhibition with extra activities, e.g., inhibition of AChE-induced Aβ aggregation, antioxidant properties, and metal-binding capacity [75,76,77,78].

Recently, we found that 2-tolylhydrazinylidene-1,3-diketones have a powerful antioxidant effect exceeding the activity of Trolox by 1.3–1.7 times [79]. Herein, we used these compounds as an antioxidant pharmacophore to create new conjugates by binding them to tacrine through an alkylene linker of various lengths. In the 2-tolylhydrazinylidene-1,3-diketone components **1a-d**, we varied the substituents at the 1,3-diketone fragment including methyl, trifluoromethyl, and phenyl residues. We studied the esterase profile of the synthesized conjugates, i.e., the inhibitory activity against AChE, BChE, and the structurally related enzyme carboxylesterase (CES, EC 3.1.1.1), whose inhibition can result in undesirable drug-drug interactions. In addition, we used quantum mechanics (QM)-assisted molecular docking to explain the observed structure-activity relationships. Moreover, we assessed the efficiency of the new compounds to displace propidium from the AChE PAS as an indicator of their potential ability to block AChE-induced Aβ aggregation. Furthermore, we determined the antioxidant activity of the conjugates in the ABTS and FRAP tests as well as their metal-chelating ability. Finally, we carried out computational predictions of the ADMET and physicochemical properties of the new conjugates.

## 2. Materials and Methods

### 2.1. Chemistry

The solvents (methanol, ethanol, chloroform, methylene chloride, hexane, acetonitrile) were obtained from AO VEKTON (St. Petersburg, Russia). Hexylamine was purchased from Alfa Aesar via Thermo Fisher Scientific (Kandel, Germany). The deuterated solvent CDCl_3_ was acquired from SOLVEX LLC (Skolkovo Innovation Center, Moscow, Russia). All solvents, chemicals, and reagents were used without purification. Melting points were determined in open capillaries on a Stuart SMP30 (Bibby Scientific Limited, Staffordshire, UK) melting point apparatus and were uncorrected. The IR spectra were recorded on a PerkinElmer Spectrum Two FT-IR spectrometer (Perkin-Elmer, Waltham, MA, USA) using the frustrated total internal reflection accessory with a diamond crystal. The ^1^H and ^19^F NMR spectra were registered on a Bruker DRX-400 spectrometer (400 or 376 MHz, respectively) or a Bruker Avance^III^ 500 spectrometer (500 or 470 MHz, respectively) (both Bruker, Karlsruhe, Germany). The ^13^C NMR spectra were recorded on a Bruker Avance^III^ 500 spectrometer (125 MHz). The internal standard was SiMe_4_ (for ^1^H and ^13^C NMR spectra) and C_6_F_6_ (for ^19^F NMR spectra, δ –162.9 ppm). The microanalyses (C, H, N) were carried out on a PerkinElmer PE 2400 series II (PerkinElmer, Waltham, MA, USA) elemental analyzer. The column chromatography was performed on Silica gel 60 (0.062–0.2 mm) (Macherey-Nagel GmbH & Co KG, Duren, Germany).

The initial 2-arylhydrazinylidene-1,3-diketones **1a,b [80]**, **1c,d** [79] and aminomethylene-tacrines **5a-c** [81] were synthesized by referring to previously published methods.

#### 2.1.1. Synthesis of Compounds **2a-c** (*General Procedure*)

A mixture of the corresponding 2-arylhydrazinylidene-1,3-diketone **1a,b** (2.3 mmol) and hexylamine (2.3 mmol) was refluxed in dry methanol for 8 h. In the case of (3E)-1,1,1-trifluoro-3-[2-(4-methylphenyl)hydrazinylidene]pentane-2,4-dione **1c**, the reaction was carried out in ethanol at room temperature. Then, the mixture was concentrated on a rotary evaporator. The residue was purified by column chromatography with the appropriate eluent, as specified below.

*(3Z)-4-(Hexylamino)-3-[(E)-(4-methylphenyl)diazenyl]pent-3-en-2-one* (**2a**). Yield 64%, orange powder, mp 36–37 °C (eluent chloroform: ethanol = 5 : 1, then chloroform). IR: ν 2926 (NH), 1645 (C=O), 1590, 1487, 1436, 1354, 1328 (N–H, C=C, N=N), 1200–1159 (C–F) cm^−1^. ^1^H NMR (400 MHz, CDCl_3_): δ 0.90–0.93, 1.34–1.36, 1.46–1.48, 1.69–1.77 (11H, all m, HNCH_2_(CH_2_)_4_CH_3_); 2.37, 2.54 (9H, all s, 3CH_3_); 3.45 (2H, unsolv. td, *J* 6.7, 4.1 Hz, HNCH_2_(CH_2_)_4_CH_3_); 7.20, 7.43 (4H, both d, *J* 8.2 Hz); 15.07 (1H, s, NH). ^13^C NMR (126 MHz, CDCl_3_): δ 13.99; 16.66; 21.10; 22.54; 26.83; 28.11; 29.29; 31.41; 44.37; 119.34; 129.18; 129.68; 136.29; 148.69; 160.98; 198.52. Anal. calcd. for C_18_H_27_N_3_O. C, 71.72; H, 9.03; N, 13.94. Found: C, 69.96; H, 8.90; N, 13.68. 

*(3Z)-3-(Hexylamino)-2-[(4-methylphenyl)diazenyl]-1-phenylbut-2-en-1-one* (**2b**). Mixture of *Z:E*–isomers-87:13. Yield 69%, orange powder, mp 54–55 °C (eluent dichloromethane). IR: ν 2924 (NH), 1630 (C=O), 1574, 1514, 1457, 1372, 1335 (N–H, C=C, N=N), 1220-1143 (C–F) cm^−1^. ^1^H NMR (400 MHz, CDCl_3_): δ 0.86–0.87 (3H, m, HNCH_2_(CH_2_)_4_CH_3_ isomer *E*); 0.91–0.94 (3H, m, HNCH_2_(CH_2_)_4_CH_3_ isomer *Z*); 1.25–1.29, 1.37–1.38, 1.50–1.53, 1.77–1.80 (8H, all m, HNCH_2_(CH_2_)_4_CH_3_, isomer *Z,E*); 2.31, 2.55 (6H, both s, 2CH_3_ isomer *Z*); 2.32, 2.63 (6H, both s, 2CH_3_ isomer *E*); 7.08–7.10, 7.12–7.13, 7.17–7.18, 7.22–7.23, 7.37–7.47, 7.54–7.56, 7.77–7.79, 7.85–7.87 (9H, all m, Ph and C_6_H_4_ isomer *Z,E*); 14.76 (1H, s, NH isomer *E*); 15.15 (1H, s, NH isomer *Z*). ^13^C NMR (125 MHz, CDCl_3_): δ 14.01; 16.64; 20.93; 21.03; 22.57; 26.91; 29.41; 30.49; 31.45; 45.09; 46.67; 114.18; 115.79; 116.17; 118.71; 119.24; 126.48; 127.19; 127.72; 128.37; 129.18; 129.61; 129.81; 129.95; 130.13; 130.21; 130.32; 131.80; 135.64; 135.96; 138.67; 139.29; 141.32; 148.14; 162.32; 192.21; 193.86. Anal. calcd. for C_23_H_29_N_3_O. C, 76.00; H, 8.04; N, 11.56. Found: C, 76.05; H, 8.22; N, 11.40.

*(3Z)-4-(Hexylamino)-3-[(E)-(4-methylphenyl)diazenyl]-1,1,1-trifluoropent-3-en-2-one* (**2c**). Yield 40%, orange powder, mp 70–72 °C (eluent chloroform, then hexane:chloroform = 1:1). IR: ν 3258 (NH), 1670 (C=O), 1591, 1558, 1511, 1459, 1365 (N–H, C=C, N=N), 1185–1148 (C–F) cm^−1^. ^1^H NMR (500 MHz, CDCl_3_): δ 0.91–0.94, 1.34–1.38, 1.46–1.50, 1.73–1.78 (11H, all m, HNCH_2_(CH_2_)_4_CH_3_); 2.38, 2.62 (6H, both s, 2CH_3_); 3.50 (2H, unsolv. td, *J* 6.8, 5.2 Hz, HNCH_2_(CH_2_)_4_CH_3_); 7.22, 7.52 (4H, both d, *J* 8.3 Hz); 14.73 (1H, s, NH). ^13^C NMR (125 MHz, CDCl_3_): δ 13.96; 16.32; 21.21; 22.50; 26.64; 28.81; 31.29; 41.88; 118.61 (q, *J* 292.2 Hz, CF_3_); 120.79; 123.23; 129.75; 138.19; 149.50; 164.11; 177.63 (q, *J* 30.4 Hz, C—CF_3_). ^19^F NMR (470 MHz, CDCl_3_): δ 92.78 (s, CF_3_). Anal. calcd. for C_18_H_24_F_3_N_3_O. C, 60.83; H, 6.81; N, 10.82. Found: C, 60.66; H, 6.94; N, 11.96.

#### 2.1.2. Synthesis of Compounds **6a-c, 7a-c, 8a-c** (*General Procedure*)

A mixture of 1,1,1-trifluoro-3-[2-(4-methylphenyl)hydrazinylidene]pentane-2,4-dione **1c** (1 mmol) and the corresponding N-(1,2,3,4-tetrahydroacridin-9-yl)alkyldiamine **5a-c** (1 mmol) in 30 mL of dry methylene chloride was stirred at room temperature for 30 min. Then 10 mL of dry methanol was added and the mixture was refluxed for 5 h. In the case of 3-[2-(4-methylphenyl)hydrazinylidene]pentane-2,4-dione **1a** and 2-[2-(4-methylphenyl)hydrazinylidene]-1-phenylbutane-1,3-dione **1b**, the reaction was carried out by refluxing in dry methanol for 8 h. Then the reaction mixture was cooled to room temperature, concentrated on a rotary evaporator, and purified by column chromatography; eluent: dichloromethane:ethanol = 20:1.

*(3Z)-3-[(E)-(4-Methylphenyl)diazenyl]-4-({2-[(1,2,3,4-tetrahydroacridin-9-yl)amino]butyl}}amino)pent-3-en-2-one* (**6a**). Yield 65%, yellow oil. IR: ν 2928 (NH), 1642 (C=O), 1581, 1562, 1499, 1415, 1352 (N–H, C=C, N=N). ^1^H NMR (500 MHz, CDCl_3_): δ 1.85-1.88 (8H, m, C^3^H_2_, C^4^H_2_ acridine and HNCH_2_(CH_2_)_2_CH_2_NH); 2.35, 2.48, 2.53 (3H, all s, 3CH_3_); 2.67, 3.06 (4H, both t, *J* 5.8 Hz, C^1^H_2_, C^2^H_2_ acridine); 3.48-3.49, 3.56 (4H, both m, HNCH_2_(CH_2_)_2_CH_2_NH); 4.02 (1H, br. s, NH tacrine); 7.13, 7.35 (4H, both d, *J* 8.2 Hz, C_6_H_4_CH_3_); 7.32–7.34, 7.54–7.57, 7.92–7.95 (4H, all m, CH_Ar_); 15.24 (1H, s, NH). ^13^C NMR (125 MHz, CDCl_3_): δ 16.90; 21.04; 22.60; 22.90; 24.79; 27.10; 27.84; 29.42; 33.72; 44.86; 48.87; 116.32; 118.60; 120.17; 122.51; 123.94; 128.51; 129.79; 130.01;130.17; 136.05; 147.02; 147.22; 150.49; 158.29; 161.93; 198.41. Anal. calcd. for C_29_H_35_N_5_O. C, 74.17; H, 7.51; N, 14.91. Found: C, 74.27; H, 7.69; N, 14.73.

*(3Z)-3-[(E)-(4-Methylphenyl)diazenyl]-4-({2-[(1,2,3,4-tetrahydroacridin-9-yl)amino]hexyl}}amino)pent-3-en-2-one* (**6b**). Yield 55%, yellow oil. IR: ν 2933 (NH), 1647 (C=O), 1642 (C=O), 1574, 1499, 1419, 1353 (N–H, C=C, N=N). ^1^H NMR (500 MHz, CDCl_3_): δ 1.49–1.50, 1.69–1.74, 1.90–1.91 (12H, all m, C^3^H_2_, C^4^H_2_ acridine and HNCH_2_(CH_2_)_4_CH_2_NH); 2.35, 2.52, 2.53 (9H, all s, 3CH_3_); 2.69, 3.07 (4H, both t, *J* 5.5 Hz, C^1^H_2_, C^2^H_2_ acridine); 3.43 (2H, td, *J* 6.6, 4.0 Hz, HNCH_2_); 3.49 (2H, t, *J* 7.1 Hz, CH_2_NH tacrine); 3.97 (1H, br. s, NH tacrine); 7.16, 7.40 (4H, both d, *J* 8.1 Hz, C_6_H_4_CH_3_); 7.31–7.34, 7.53–7.57, 7.92–7.94 (4H, all m, CH_Ar_); 15.12 (1H, s, NH). ^13^C NMR (125 MHz, CDCl_3_): δ 16.72; 21.05; 22.69; 22.99; 24.77; 26.63; 26.98; 29.32; 31.61; 33.84; 44.47; 49.26; 115.93; 119.03; 120.14; 122.66; 123.71; 128.39; 128.57; 129.51; 129.70; 136.21; 147.18; 148.16; 150.70; 158.26; 161.26; 198.43. Anal. calcd. for C_31_H_39_N_5_O. C, 74.81; H, 7.90; N, 14.07. Found: C, 74.38; H, 8.24; N, 13.88.

*(3Z)-3-[(E)-(4-Methylphenyl)diazenyl]-4-({2-[(1,2,3,4-tetrahydroacridin-9-yl)amino]octyl}}amino)pent-3-en-2-one* (**6c**). Yield 45%, yellow oil. IR: ν 2926 (NH), 1635 (C=O), 1583, 1516, 1456, 1415, 1353 (N–H, C=C, N=N). ^1^H NMR (500 MHz, CDCl_3_): δ 1.41–1.47, 1.74–1.80, 1.88–1.93, 1.91–1.92 (16H, all m, C^3^H_2_, C^4^H_2_ acridine and HNCH_2_(CH_2_)_6_CH_2_NH); 2.33, 2.53 (9H, both s, 3CH_3_); 2.59, 3.28 (4H, both t, *J* 5.3 Hz, C^1^H_2_, C^2^H_2_ acridine); 3.46, 3.80 (4H, both br.s, HNCH_2_(CH_2_)_6_CH_2_NH); 5.27 (1H, br. s, NH tacrine); 7.17, 7.41 (4H, both d, *J* 8.0 Hz, C_6_H_4_CH_3_); 7.41–7.43, 7.66–7.69, 8.10–8.11, 8.43–8.44 (4H, all m, CH_Ar_); 15.09 (1H, s, NH).^13^C NMR (125 MHz, CDCl_3_): δ 16.70; 21.07; 21.22; 22.19; 23.92; 26.63; 26.96; 28.06; 29.01; 29.09; 29.25; 29.95; 31.26; 44.35; 48.77; 112.13; 116.21; 117.01; 119.19;. 123.25; 123.75; 124.64; 129.29; 129.65; 130.17; 131.10; 136.28; 148.45; 154.01; 161.11; 198.48. Anal. calcd. for C_33_H_43_N_5_O. C, 75.39; H, 8.24; N, 13.32. Found: C, 75.43; H, 8.17; N, 13.69.

*(2Z)-2-[(E)-(4-Methylphenyl)diazenyl]-1-phenyl-3-({2-[(1,2,3,4-tetrahydroacridin-9-yl)amino]butyl}amino)but-2-en-1-one* (**7a**). Yield 68%, yellow oil. IR: ν 2930 (NH), 1626 (C=O), 1578, 1499, 1416, 1335 (N–H, C=C, N=N). ^1^H NMR (500 MHz, CDCl_3_): δ 1.88–1.91 (8H, m, C^3^H_2_, C^4^H_2_ acridine and HNCH_2_(CH_2_)_2_CH_2_NH); 2.30, 2.48 (6H, both s, 2CH_3_); 2.70, 3.08 (4H, both t, *J* 5.8 Hz, C^1^H_2_, C^2^H_2_ acridine); 3.57-3.61 (4H, m, HNCH_2_(CH_2_)_2_CH_2_NH); 3.99 (1H, br. s, NH tacrine); 7.03, 7.10 (4H, both d, *J* 8.1 Hz, C_6_H_4_CH_3_); 7.34–7.37, 7.39–7.42, 7.74–7.48 (5H, all m, C_6_H_5_); 7.56–7.59, 7.81–7.82, 7.94–7.96 (4H, all m, CH_Ar_); 15.37 (1H, s, NH).^13^C NMR (125 MHz, CDCl_3_): δ 16.91; 20.99; 22.68; 22.94; 24.83; 27.23; 29.52; 33.96; 45.69; 48.97; 116.51; 118.40; 120.33; 122.51; 123.87; 127.29; 128.36; 128.80; 129.72 (2C); 130.02; 130.26; 130.68; 135.65; 140.67; 146.56; 147.37; 150.33; 158.56; 163.26; 193.60. Anal. calcd. for C_34_H_37_N_5_O. C, 76.80; H, 7.01; N, 13.17. Found: C, 76.65; H, 7.15; N, 13.29.

*(2Z)-2-[(E)-(4-Methylphenyl)diazenyl]-1-phenyl-3-({2-[(1,2,3,4-tetrahydroacridin-9-yl)amino]hexyl}amino)but-2-en-1-one* (**7b**). Yield 57%, yellow oil. IR: ν 2930, 2858 (NH), 1626 (C=O), 1578, 1499, 1366, 1335 (N–H, C=C, N=N). ^1^H NMR (500 MHz, CDCl_3_): δ 1.51–1.55, 1.69–1.75, 1.76–1.80, 1.90–1.91 (12H, all m, C^3^H_2_, C^4^H_2_ acridine and HNCH_2_(CH_2_)_4_CH_2_NH); 2.29, 2.51 (6H, both s, 2CH_3_); 2.69, 3.07 (4H, both t, *J* 5.7 Hz, C^1^H_2_, C^2^H_2_ acridine); 3.48-3.53 (4H, m, HNCH_2_(CH_2_)_4_CH_2_NH); 4.02 (1H, br. s, NH tacrine); 7.05, 7.14 (4H, both d, *J* 8.1 Hz, C_6_H_4_CH_3_); 7.32–7.35, 7.37–7.42; 7.54–7.57 (5H, all m, C_6_H_5_); 7.54–7.57, 7.78–7.79, 7.93–7.95 (4H, all m, CH_Ar_); 15.21 (1H, s, NH).^13^C NMR (125 MHz, CDCl_3_): δ 16.72; 21.00; 22.64; 22.95; 24.74; 26.66; 27.04; 29.42; 31.68; 33.74; 45.19; 49.27; 115.83; 118.91; 120.05; 122.71; 123.72; 127.23 (2C); 128.45; 129.46; 129.63; 129.84; 130.21; 130.48; 135.90; 141.04; 147.05; 147.59; 150.78; 158.17; 162.60; 193.76. Anal. calcd. for C_36_H_41_N_5_O. C, 77.25; H, 7.38; N, 12.51. Found: C, 77.55; H, 7.68; N, 12.23.

*(2Z)-2-[(E)-(4-Methylphenyl)diazenyl]-1-phenyl-3-({2-[(1,2,3,4-tetrahydroacridin-9-yl)amino]octyl}amino)but-2-en-1-one* (**7c**). Yield 47%, yellow oil. IR: ν 2927, 2855 (NH), 1628 (C=O), 1576, 1514, 1501, 1366, 1335 (N–H, C=C, N=N). ^1^H NMR (500 MHz, CDCl_3_): δ 1.41–1.51, 1.68-1.72, 1.74-1.79, 1.90-1.91 (16H, all m, C^3^H_2_, C^4^H_2_ acridine and HNCH_2_(CH_2_)_6_CH_2_NH); 2.28, 2.53 (6H, both s, 2CH_3_); 2.65, 3.12 (4H, both t, *J* 5.3 Hz, C^1^H_2_, C^2^H_2_ acridine); 3.51 (2H, td, *J* 6.7, 4.1 Hz, HNCH_2_); 3.56 (2H, t, *J* 7.1 Hz, CH_2_NH tacrine); 3.97 (1H, br s, NH tacrine); 4.27 (1H, br. s, NH tacrine); 7.07, 7.16 (4H, both d, *J* 8.1 Hz, C_6_H_4_CH_3_); 7.34–7.41, 7.44–7.46 (5H, all m, C_6_H_5_); 7.56–7.60, 7.77–7.79, 7.97–7.99 (4H, all m, CH_Ar_); 15.17 (1H, s, NH).^13^C NMR (125 MHz, CDCl_3_): δ 16.65; 20.99; 22.24; 22.73; 24.44; 26.76; 29.09; 29.19; 29.35; 31.56; 45.07; 49.21; 114.66; 119.10; 123.08; 123.88; 126.94; 127.20; 128.35; 129.14; 129.26; 129.58; 130.17; 130.28; 130.40; 135.95; 139.30; 141.15; 145.50; 147.92; 151.73; 156.82; 162.40; 193.80. Anal. calcd. for C_38_H_45_N_5_O. C, 77.65; H, 7.72; N, 11.91. Found: C, 77.45; H, 7.92; N, 11.74.

*(3Z)-3-[(E)-(4-Methylphenyl)diazenyl]-4-({2-[(1,2,3,4-tetrahydroacridin-9-yl)amino]butyl}amino-1,1,1-trifluoropent-3-en-2-one* (**8a**). Yield 60%, crystalizing oil. IR: ν 2942, 2868 (NH), 1671 (C=O), 1585, 1562, 1496, 1368 (N–H, C=C, N=N), 1220–1143 (C–F) cm^−1^. ^1^H NMR (500 MHz, CDCl_3_): δ 1.85–1.90 (8H, m, C^3^H_2_, C^4^H_2_ acridine and HNCH_2_(CH_2_)_2_CH_2_NH); 2.37, 2.59 (6H, both s, 2CH_3_); 2.69, 3.06 (4H, both t, *J* 6.0 Hz, C^1^H_2_, C^2^H_2_ acridine); 3.52–3.54 (4H, m, HNCH_2_(CH_2_)_2_CH_2_NH); 3.88 (1H, br. s, NH tacrine); 7.17, 7.47 (4H, both d, *J* 8.2 Hz, C_6_H_4_CH_3_); 7.33–7.36, 7.46–7.48, 7.55–7.58; 7.89–7.93 (4H, all m, CH_Ar_); 14.88 (1H, s, NH). ^13^C NMR (125 MHz, CDCl_3_): δ 16.32; 21.20; 22.69; 22.92; 24.86; 26.48; 29.15; 34.01; 43.62; 48.67; 116.87; 118.50 (q, *J* 292.1 Hz, CF_3_); 120.44; 120.61; 122.32; 123.41; 123.99; 128.38; 128.94; 129.82; 138.34; 147.43; 149.10; 150.06; 158.71; 164.15; 177.67 (q, *J* 30.4 Hz, C—CF_3_). ^19^F NMR (470 MHz, CDCl_3_): δ 92.68 (s, CF_3_). Anal. calcd. for C_29_H_32_F_3_N_5_O. C, 66.52; H, 6.16; N, 13.38. Found: C, 66.30; H, 6.36; N, 13.02.

*Crystallographic data for compound***8a**. The X-ray studies were performed on an Xcalibur 3 CCD (Oxford Diffraction Ltd., Abingdon, UK) diffractometer with a graphite monochromator, ω scanning with 1° step, λ(MoKα) 0.71073 Å radiation, T 295(2) K. An empirical absorption correction was applied. Using Olex2 [82], the structure was solved with the ShelXT [83] structure solution program using Direct Methods and refined with the ShelXL [84] refinement package using Least Squares minimization. All non-hydrogen atoms were refined in the anisotropic approximation; H-atoms at the C-H bonds were refined in the “rider” model with dependent displacement parameters. An empirical absorption correction was carried out through spherical harmonics, implemented in the SCALE3 ABSPACK scaling algorithm by a program “CrysAlisPro 1.171.41.123a” (Rigaku Oxford Diffraction, 2022).

The suitable orange single crystals of compound **8a** were obtained by slow crystallization from acetonitrile. Main crystallographic data for **8a**: C_29_H_32_F_3_N_5_O, M 523.59, space group P2_1_/n, monoclinic, *a* 9.6290(8), *b* 24.6933(16), *c* 11.4045(8) Å; *β* 103.973(7)°; *V* 2631.4(3) Å^3^; *Z* 4; *D*_calc_ 1.322 g∙cm^−3^; *μ* 0.097 mm^−1^; 370 refinement parameters; 22,006 reflections measured, 7024 [R_int_ = 0.0580, R_sigma_ = 0.0626] unique reflections which were used in all calculations. The final R_1_ = 0.0843, wR_2_ = 0.2237 [I ≥ 2σ (I)], R_1_ = 0.1441, wR_2_ = 0.2836 [all data]. CCDC 2165596 contains the Appendix A for this compound.

*(3Z)-3-[(E)-(4-Methylphenyl)diazenyl]-4-({2-[(1,2,3,4-tetrahydroacridin-9-yl)amino]hexyl}amino-1,1,1-trifluoropent-3-en-2-one* (**8b**). Yield 48%, yellow oil. IR: ν 2931 (NH), 1663 (C=O), 1591, 1562, 1497, 1421, 1382 (N–H, C=C, N=N), 1169-1143 (C–F) cm^−1^. ^1^H NMR (500 MHz, CDCl_3_) δ 1.55–1.56, 1.78–1.88 (12H, all m, C^3^H_2_, C^4^H_2_ acridine and HNCH_2_(CH_2_)_4_CH_2_NH); 2.36, 2.60 (6H, both s, 2CH_3_); 2.62, 3.19 (4H, both t, *J* 5.6 Hz, C^1^H_2_, C^2^H_2_ acridine); 3.52 (2H, td, *J* 6.6, 5.2 Hz, HNCH_2_); 3.75 (2H, t, *J* 6.9 Hz, CH_2_NH tacrine); 5.07 (1H, br. s, NH tacrine); 7.18, 7.48 (4H, both d, *J* 8.3 Hz, C_6_H_4_CH_3_); 7.35–7.38, 7.60–7.63, 8.04–8.05, 8.26–8.28 (4H, all m, CH_Ar_); 14.75 (1H, s, NH).^13^C NMR (125 MHz, CDCl_3_) δ 16.36; 21.20; 21.45; 22.32; 24.23; 26.38; 26.67; 28.75; 29.67; 31.22; 43.70; 48.57; 112.98; 117.64; 118.53 (q, *J* 291.8 Hz, CF_3_); 120.68; 123.29; 123.52; 124.09; 124.51; 129.76; 130.56; 138.30; 142.48; 149.34; 153.36; 154.33; 164.16; 177.59 (q, *J* 30.4 Hz, C—CF_3_). ^19^F NMR (470 MHz, CDCl_3_): δ 92.73 (s, CF_3_). Anal. calcd. for C_31_H_36_F_3_N_5_O. C, 67.49; H, 6.58; N, 12.70. Found: C, 67.39; H, 6.35; N, 12.42.

*(3Z)-3-[(E)-(4-Methylphenyl)diazenyl]-4-({2-[(1,2,3,4-tetrahydroacridin-9-yl)amino]octyl}amino-1,1,1-trifluoropent-3-en-2-one* (**8c**). Yield 42%, yellow oil. IR: ν 2927 (NH), 1663 (C=O), 1592, 1562, 1498, 1421, 1382 (N–H, C=C, N=N), 1169-1144 (C–F) cm^−1^. ^1^H NMR (500 MHz, CDCl_3_) δ 1.38-1.48, 1.63–1.67, 1.72–1.75, 1.91–1.92 (16H, all m, C^2^H_2_, C^3^H_2_ acridine + HNCH_2_(CH_2_)_6_CH_2_NH); 2.36, 2.61 (6H, both s, 2CH_3_); 2.70, 3.06 (4H, both t, *J* 5.3 Hz, C^1^H_2_, C^4^H_2_ acridine); 3.45-3.51 (4H, m, HNCH_2_(CH_2_)_6_CH_2_NH); 3.93 (1H, br. s, NH tacrine); 7.20, 7.50 (4H, both d, *J* 8.2 Hz, C_6_H_4_CH_3_); 7.32–7.35, 7.53–7.56, 7.90–7.94 (4H, all m, CH_Ar_); 14.73 (1H, s, NNH).^13^C NMR (125 MHz, CDCl_3_): δ 16.34; 21.20; 22.73; 23.01; 24.75; 26.81; 26.86; 28.82; 29.04; 29.19; 31.70; 33.91; 43.81; 49.38; 115.83; 118.57 (q, *J* 292.1 Hz, CF_3_); 120.15; 120.76; 122.77; 123.28; 123.61; 128.34; 128.61; 129.75; 138.24; 147.31; 149.46; 150.76; 158.34; 164.11; 177.66 (q, *J* 30.8 Hz, C—CF_3_). ^19^F NMR (470 MHz, CDCl_3_): δ 92.75 (s, CF_3_). Anal. calcd. for C_33_H_40_F_3_N_5_O. C, 68.37; H, 6.96; N, 12.08. Found: C, 68.54; H, 7.05; N, 12.13.

### 2.2. Biological Testing

#### 2.2.1. Enzymatic Assays

##### AChE, BChE and CES Inhibition 

Human erythrocyte AChE, equine serum BChE, porcine liver CES, acetylthiocholine iodide (ATCh), butyrylthiocholine iodide (BTCh), 5,5′-dithio-bis-(2-nitrobenzoic acid) (DTNB), 4-nitrophenyl acetate (4-NPA), tacrine were purchased from Sigma-Aldrich (St. Louis, MO, USA). AChE and BChE activities were measured by the colorimetric method of Ellman (λ 412 nm). The assay solution consisted of 0.1 M K/Na phosphate buffer pH 7.5, 25 °C, 0.33 mM DTNB, 0.02 unit/mL AChE or BChE, and 1 mM substrate (ATCh or BTCh, respectively). Reagent blanks consisted of reaction mixtures without substrates.

The activity of CES was determined spectrophotometrically by the release of 4-nitrophenol at 405 nm in 0.1 M K/Na phosphate buffer pH 8.0, 25 °C. Final enzyme and substrate (4-nitrophenyl acetate) concentrations were 0.02 unit/mL and 1 mM, respectively. Assays were carried out with a blank containing all constituents except porcine CES to assess non-enzymatic hydrolysis. Measurements were performed with a FLUOStar Optima microplate reader (BMG Labtech, Ortenberg, Germany). Compounds were dissolved in DMSO; the incubation mixture contained 2% (*v*/*v*) solvent. The primary evaluation of the inhibitory activity of the compounds was performed by determining the degree of the enzyme inhibition at a compound concentration of 20 µM. For this, a sample of the corresponding enzyme was incubated with the test compound for 5 min; then the enzyme residual activity was determined. Each experiment was performed in triplicate. Compounds inhibiting the enzyme by more than 30% were then selected for determination of IC_50_ values (the inhibitor concentration resulting in 50% inhibition of control enzyme activity). Compounds (eight concentrations ranging between 1 × 10^−11^ and 1 × 10^−4^ M were used to achieve 20 to 80% inhibition) were incubated with each enzyme for 5 min at 25 °C (for temperature equilibration). Substrate was then added and residual enzyme activity relative to an inhibitor-free control was measured using a FLUOStar Optima microplate reader.

##### Kinetic Study of AChE and BChE Inhibition. Determination of Inhibition Mechanism and Steady-State Inhibition Constants

Mechanisms of human erythrocyte AChE and equine serum BChE inhibition were assessed via a thorough analysis of enzyme kinetics. Residual activity was measured following 5 min incubation at 25 °C with three increasing concentrations of inhibitor and six decreasing substrate concentrations. Inhibition constants *K*_i_ (competitive component) and α*K*_i_ (noncompetitive component) were determined by linear regression of 1/V versus 1/[S] double-reciprocal (Lineweaver–Burk) plots.

#### 2.2.2. Propidium Displacement from *Ee*AChE PAS

Propidium iodide, donepezil, and Tris were purchased from Sigma-Aldrich (St. Louis, MO, USA). The ability of the test compounds to competitively displace propidium, a selective ligand of the PAS of AChE, was evaluated by a fluorescence method [85]. *Electrophorus electricus* (*Ee*AChE) (electric eel, type VI-S, lyophilized powder, Sigma-Aldrich, St. Louis, MO, USA) was used owing to its high degree of purification, high activity, and lower cost than human AChE (hAChE). The applicability of this enzyme has been substantiated earlier [67]. The fluorescence intensity of propidium iodide bound with AChE increases several times; decreasing fluorescence intensity of the bound propidium in the presence of the test compounds shows their ability to bind to the PAS of AChE. To determine the degree of displacement (% displacement) of propidium from the PAS of AChE, *Ee*AChE (final concentration, 7 μM) was incubated with the test compound at a concentration of 20 μM in 1 mM Tris-HCl buffer pH 8.0, 25 °C, for 15 min. Then, propidium iodide solution (final concentration, 8 μM) was added, the samples were incubated for 15 min, and the fluorescence spectrum (530 nm (excitation) and 600 nm (emission)) was taken. Donepezil and tacrine were used as reference compounds. The blank contained propidium iodide of the same concentration in 1 mM Tris-HCl buffer, pH 8.0 at 25 °C. The measurements were carried out in triplicate on a FLUOStar Optima microplate reader (LabTech, Ortenberg, Germany), and the results were calculated by the following formula:% Displacement = 100 − (IF_AChE + Propidium + inhibitor_ / IF_AChE + Propidium_) × 100 (1)
where IF_AChE + Propidium_ is the fluorescence intensity of the propidium associated with AChE in the absence of the test compound (taken as 100%), and IF_AChE + Propidium + inhibitor_ is the fluorescence intensity of the propidium associated with AChE in the presence of the test compound.

#### 2.2.3. ABTS Radical Cation Scavenging Activity Assay

Radical scavenging activity of the compounds was evaluated by the ABTS radical cation (ABTS^•+^) scavenging assay showing the ability of the compounds to decolorize the ABTS^•+^ solution [86] with some modifications [87]. Trolox was used as the antioxidant standard; ascorbic acid was used as the comparison compound. All tested compounds and standards were dissolved in DMSO.

ABTS (2,2ʹ-azinobis-(3-ethylbenzothiazoline-6-sulfonic acid)) was purchased from Tokyo Chemical Industry Co., Ltd. (Tokyo, Japan). Potassium persulfate (dipotassium peroxydisulfate), Trolox (6-hydroxy-2,5,7,8-tetramethylchroman-2-carboxylic acid), ascorbic acid, DMSO and HPLC-grade ethanol were obtained from Sigma-Aldrich Chemical Co. (St. Louis, MO, USA). Aqueous solutions were prepared using deionized water.

The solution of ABTS^•+^ was produced by mixing 7 mM ABTS aqueous solution with 2.45 mM potassium persulfate aqueous solution in equal quantities and allowing them to react for 12–16 h at room temperature in the dark. Radical scavenging capacity of the compounds was analyzed by mixing 10 µL of compound with 240 µL of ABTS^•+^ working solution in ethanol (100 µM final concentration), and after 1 h of incubation at 30 °C the decrease in absorbance was measured spectrophotometrically at 734 nm using a Bio-Rad xMark microplate UV/VIS spectrophotometer (Bio-Rad, Hercules, CA, USA). The compounds were tested in the concentration range 5 × 10^−7^–1 × 10^−4^ M. Ethanol blanks were run in each assay. Values were obtained from five replicates of each sample and three independent experiments.

Antioxidant activity was reported as Trolox equivalent antioxidant capacity (TEAC values), consisting of the ratio between the slopes obtained from the linear correlation for concentrations of the tested compounds and Trolox with absorbance of ABTS radical. For the compounds, we also determined IC_50_ values (compound concentration (μM) required for 50% reduction of the ABTS radical). The calculations were carried out using Origin 6.1 for Windows (OriginLab, Northampton, MA, USA). Results are presented as mean ± SEM calculated using GraphPad Prism version 6.05 for Windows, GraphPad Software (San Diego, CA, USA).

#### 2.2.4. Ferric Reducing Antioxidant Power (FRAP) Assay

The FRAP (Ferric Reducing Antioxidant Power) assay measures the ability of antioxidants to reduce the ferric 2,4,6-tripyridyl-s-triazine complex [Fe(TPTZ)_2_]^3+^ to the intensely blue-colored ferrous complex [Fe(TPTZ)_2_]^2+^ with an absorption maximum at λ = 593 nm [88,89]. The reducing ability of a compound is an indicator of its potential antioxidant activity [90].

The ferric reducing ability of the compounds was determined by a previously described method [89] as a microplate-adapted version described in [42]. 2,4,6-tris(pyridin-2-yl)-1,3,5-triazine (TPTZ), FeCl_3_·6H_2_O, Trolox, ascorbic acid and DMSO were obtained from Sigma-Aldrich Chemical Co (St. Louis, MO, USA). The FRAP reagent was prepared by mixing acetate buffer (0.3 M, pH 3.6), TPTZ (10 mM in 40 mM HCl) and FeCl_3_·6H_2_O (20 mM in distilled water) in a ratio of 10:1:1 immediately before use. Compounds were dissolved in DMSO and tested in the concentration range of 1×10^−6^–1×10^−4^ M. The solvent content in the reaction mixture was 4% (*v/v*). The test compounds (10 µL) were added to the FRAP reagent solution (240 µL) and mixed thoroughly. The reaction was carried out at 37 °C in the dark, the incubation time was 1 h. The absorbance at 600 nm was monitored spectrophotometrically by a FLUOStar OPTIMA microplate reader (BMG Labtech, Germany) at 37 °C. Trolox was used as a standard antioxidant, ascorbic acid as a reference compound. Values were obtained from four replicates of each sample and three independent experiments.

The ferric reducing ability of compounds was expressed as TE units (antioxidant activity in Trolox equivalent) with the values calculated as the ratio of the concentrations of Trolox and the test compound resulting in the same effect.

#### 2.2.5. Metal-chelating Properties of Compounds **6a**, **7a** and **8a**

The complexing studies were made in acetonitrile at 25 °C using a UV–vis spectrophotometer Shimadzu UV-2600 (Shimadzu Corporation, Kyoto, Japan) with wavelength ranging from 190 to 600 nm. Solutions (200 µM in acetonitrile) of the following metals compounds were prepared in volumetric flasks: FeCl_2_·4H_2_O (99%, Acros Organics by Thermo Fisher Scientific (Kandel, Germany)), CuCl_2_ (98%, Alfa Aesar by Thermo Fisher Scientific (Kandel, Germany)), or Zn(NO_3_)_2_·6H_2_O (98%, Alfa Aesar by Thermo Fisher Scientific (Kandel, Germany). Solutions of the test compounds were also prepared in acetonitrile at 400 µM concentrations. To a mixture of 0.5 mL test compound solution (40 μM final concentration) and 3.5 mL acetonitrile, 1 mL of the metal solution (CuCl_2_, FeCl_2_·4H_2_O, or Zn(NO_3_)_2_·6H_2_O; 40 µM final concentration) was added. The solution was incubated at 25 °C for 30 min and then the absorption spectra were recorded at 25 °C in a 1 cm quartz cell. The control was prepared by mixing 0.5 mL tested compound solution and 4.5 mL acetonitrile.

#### 2.2.6. Molecular Modeling Studies

##### QM analysis of the Structures

Estimations of p*K*_a_ values were performed with the Calculator Plugins of MarvinSketch 21.14.0, ChemAxon (http://www.chemaxon.com, accessed on 19 September 2021). Because the p*K*_a_ values of the tacrine fragment for all considered compounds was estimated as 8.89, all conjugates were used for all further calculations with a protonated endocyclic nitrogen atom of the tacrine fragment.

For all considered compounds an array of possible tautomers was also generated with the Calculator Plugins of MarvinSketch 21.14.0. A conformational search was performed with TorsiFlex v. 2021.3 [91,92], for all torsions of the hydrazone-diketone fragment, with connectivity of the mobile hydrogen skipped. The following main parameters of TorsiFlex were used: 10,000 of steps of a stochastic algorithm (increased to 20,000 and 50,000 when an additional search was needed), HF/3-21G for low level and B3LYP/6-31G* for high level calculations.

QM optimization of the generated structures, as well as proton transfer pathway calculations (optimization, transition state (TS) search and intrinsic reaction path calculations (IRC)) were performed with Gaussian 16 [93] using a DFT method (B3LYP/6-31G*).

##### Molecular Docking

The X-ray structure of human AChE co-crystallized with donepezil (PDB ID 4EY7 [94]) after removal of water molecules and other molecules, and an optimized X-ray structure of human BChE (PDB ID 1P0I [95,96]) were used for molecular docking. Structures of the ligands in the most stable configuration after QM optimization were used. Partial atomic charges on ligand atoms were assigned from QM data according to the Löwdin scheme [97].

Molecular docking was performed with AutoDock 4.2.6 software [98]. The grid box for docking included the entire active site gorge of AChE (22.5Å × 22.5Å × 22.5Å grid box dimensions) and BChE (15Å × 20.25Å × 18Å grid box dimensions) with a grid spacing of 0.375 Å. The main Lamarckian Genetic Algorithm (LGA) [99] parameters were 256 runs, 25 × 10^6^ evaluations, 27 × 10^4^ generations, and a population size of 3000.

Figures were prepared with PyMol (www.pymol.org, accessed on 21 July 2016).

#### 2.2.7. Prediction of ADMET and Physicochemical Profiles

Lipophilicity (LogP_ow_) and aqueous solubility (pS) were estimated by the ALogPS 3.0 neural network model implemented in the OCHEM platform [100]. Human intestinal absorption (HIA) [101], blood–brain barrier distribution/permeability (LogBB) [102,103], and hERG-mediated cardiac toxicity risk (channel affinity p*K_i_* and inhibitory activity pIC_50_) [104] were estimated using the integrated online service for the prediction of ADMET properties [105]. This service implements predictive QSAR models based on accurate and representative training sets, fragmental descriptors, and artificial neural networks. The quantitative estimate of drug-likeness (QED) values [106] were calculated using RDKit version 2021.09.2 software [107].

## 3. Results and Discussion

### 3.1. Chemistry

First, the interaction of 2-tolylhydrazinylidene-1,3-diketones **1a-d** with hexylamine was studied as a model reaction for the synthesis of conjugates with aminomethylene-modified tacrine. It was found that 2-tolylhydrazinylidene-substituted acetyl- and benzoylacetones **1a,b** reacted with hexylamine at the acetyl moiety in refluxing methanol to give products **2a,b** in good yields (Figure 1). Note that the reaction of the benzoylacetone derivative **1b** bearing non-equivalent carbonyl centers proceeded chemoselectively to produce the only product **2b**. The regioisomeric structure of compound **2b** was established by ^13^C NMR spectroscopy. The signal of the carbonyl carbon atom in the ^13^C NMR spectrum was observed at δ 193 ppm, typical of the benzoyl fragment. The resonating signal of the acetyl group carbon atom in product **2a** was observed in the downfield region at δ 198 ppm [108].

The reaction of 1,1,1-trifluoro-3-[2-(4-methylphenyl)hydrazinylidene]pentane-2,4- dione **1c** and hexylamine occurred less selectively. This was confirmed by the formation of a mixture of products from which 1,1,1-trifluoro-4-hexylimine-3-(2-[4-methylphenyl)hydrazinylidene]pentan-2-one **2c** was isolated in moderate yield (Figure 1). Its regioisomeric structure was confirmed by ^13^C NMR spectroscopy. In the ^13^C NMR spectrum of **2c**, the signal of the carbonyl atom at the CF_3_ group was observed as a quartet in the same region at δ ~ 177 ppm as for the initial diketone **1c** [109]. We have shown earlier the preference in the condensation of trifluoromethyl-containing 2-arylhydrazinylidene-1,3-diketones at the carbonyl group of the non-fluorinated substituent for reactions with methylamine [110]. However, the mass spectrum of the reaction mixture had molecular ion peaks of 2,2,2-trifluoro-N-hexylacetamide **3** (*m/z* [C_8_H_14_F_3_NO]^+^ = 198) and 1-[2-(4-methylphenyl)hydrazinylidene]propan-2-one **4a** (*m/z* [C_10_H_12_N_2_O]^+^ = 176), formed as a result of competitive condensation of hexylamine at the carbonyl group with CF_3_ substituent and subsequent cleavage of intermediate **A**. This side reaction resulted in a moderate yield of product **2c.**

In contrast, the reaction of the 4,4,4-trifluoro-2-[2-(4-methylphenyl)hydrazinylidene]-1-phenylbutane-1,3-dione **1d** with hexylamine in refluxing methanol or at room temperature led to a mixture of products. According to the GC/MS, there were predominant peaks corresponding to molecular ions of 2,2,2-trifluoro-N-hexylacetamide **3** and 2-[2-(4-methylphenyl)hydrazinylidene]-1-phenylethan-1-one **4b** (*m/z* [C_15_H_14_N_2_O]^+^ = 239). In this case, the addition of hexylamine at the trifluoroacyl group of 1,3-diketone **1d** to form intermediate **A** and its subsequent cleavage to amide **3** and ketone **4b** were apparently to become preferable (Figure 1). It should be noted that at room temperature incomplete conversion of the initial diketone **1d** along with the formation of cleavage products **3** and **4b** was observed, and at a temperature of 0–5 °C there were no noticeable changes in the initial reagents.

Then, it was found that 4-tolylhydrazinylidene-1,3-diketones **1a-c** react with aminomethylene tacrine derivatives **5a-c** (synthesized according to the methodology [81]) to give conjugates **6a-c, 7a-c, 8a-c** with various substituents in the diketone moiety and the length of the methylene linker (Figure 2). The best yields of products **6a-c** and **7a-c** were achieved in the reactions of non-fluorinated 1,3-diketones **1a** and **1b** in dry methanol under reflux for 8 h. The reaction of the trifluoromethyl-containing analogue **1c** was carried out in a mixture of dry methylene chloride and dry methanol (3: 1) under reflux for 5 h. Note that the reaction in dry methanol or ethanol even at room temperature led to acidic cleavage of the starting 2-tolylhydrazinylidene-1,3-diketone **1c** and significantly reduced the yields of the target conjugates **8a-c**.

The regioisomeric structure of compounds **6a-c**, **7a-c**, **8a-c** was established by ^13^C NMR spectroscopy. In the spectra of acetylacetone derivatives **6a-c**, a characteristic signal of the carbonyl carbon atom of the acetyl group was observed at δ 198 ppm. In the case of benzoylacetone derivatives **7a-c**, the carbonyl carbon atom of the benzoyl fragment resonated at δ 193 ppm, while the spectra of trifluoroacetylacetone derivatives **8a-c** were characterized by a quartet signal of the trifluoroacetyl carbon atom at δ 177 ppm. It can be concluded that all compounds **6a-c**, **7a-c**, **8a-c** were formed by condensation of 1,3-diketones **1a-c** with tacrines **5a-c** at the acetyl fragment similarly to the reactions with hexylamine.

It should be noted that the reaction of diketone **1d** with tacrines **5a-c** was not effective despite attempts to vary the conditions.

The synthesized compounds **2a-c, 6a-c, 7a-c,** and **8a****-c** have a hydrazone-diketone fragment with a mobile hydrogen atom. They can be characterized by prototropic imino-amine, azo-hydrazone and keto-enol tautomerism with the existence of four tautomeric forms **AAK**, **HIK**, **AIE,** and **AIK**, including *Z,E*-isomers for the first three forms (Figure 3). All tautomers except **AIK** are stabilized by an intramolecular H-bond.

The ^1^H NMR spectra of compounds **2a-c, 6a-c, 7a-c**, and **8a-c** in CDCl_3_ did not contain the CH proton signal of the **AIK** form. Instead, a low-field signal of the proton of the HN- or HO-group was observed at δ 14.7–15.4 ppm and, as already mentioned, all ^13^C NMR spectra contained low-field signals of carbonyl carbon atoms. These spectral data allowed us to exclude the **AIK** and **AIE** tautomers from consideration.

The choice between tautomers **AAK** and **HIK** was made using ^1^H NMR spectral data and our previous experience. We have earlier shown that 4-(*N*-methyl)amino-1,1,1-trifluoro-3-phenylazopent-3-en-2-one [110] exists in the solid state as an *Z*-azo-amino-ketone tautomer (***Z*-AAK**) according to X-ray diffraction data, and upon dissolution in chloroform transforms into the *Z*-hydrazo-keto-imine tautomer (***Z*-HIK**), given that in its ^1^H NMR spectrum the signal of the *N*-methyl group was observed as a singlet at δ 2.92 ppm. In contrast, in the ^1^H NMR spectra of compounds **2a****-c**, **6a****-c**, **7a****-c**, and **8a****-c**, the signals of the N-CH_2_ groups were observed as a triplet of doublets at δ 3.4–3.5 ppm due to interaction with the neighboring NH proton.

The ^1^H and ^13^C NMR spectra of products **2a,c**, **6a****-c**, **7a****-c**, **8a****-c** in CDCl_3_ contained one set of signals. The spectra of compound **2b** containing benzoyl and *N*-hexylamine substituents were characterized by the presence of a second set of signals, and its ^1^H NMR spectrum exhibited two triplets of doublets of a HN-CH_2_ group at δ 3.18 and 3.52 ppm, apparently due to the existence of compounds **2b** in the form of *Z,E*-isomers in a ratio of 87: 13.

Based on the analysis performed, we believe that in a CDCl_3_ solution, compounds **2a-c**, **6a-c**, **7a-c**, and **8a-c** predominantly exist in the ***Z*-AAK** form in contrast to the *N*-methyl analog [110] (Figure 3).

We obtained monocrystals for compound **8a** by slow crystallization from acetonitrile. The XRD analysis showed that conjugate **8a** exists in the solid state also as ***Z*-AAK** (Figure 1) with formation of the intramolecular hydrogen bond between hydrogen H3 of NH group and nitrogen N2 of arylazo moiety (the distance H3···N2 is 1.650 Å) similarly to the *N*-methyl analog [110]. The molecule of compound **8a** has an almost flat structure because 4-amino-1,1,1-trifluoro-3-(tolyldiazenyl)pent-3-en-2-one moiety and tacrine core are located in one plane. However, the butyl spacer C18C17C16C15 is characterized by distortions with 1.293 Å (for C18) and 1.131 Å (for C15) deviations from the common plane. The distance between nitrogens N3···N4 connected with butyl linker is 5.304 Å.

To assess computationally the relative stability of tautomers and conformers of the compounds **2a-c**, **6a-c, 7a-c**, and **8a-c**, instead of manual generation of configurations of interest, as we have formerly done [79,87,111], we resorted to formal generation of tautomers with ChemAxon MarvinSketch. Next, geometries of all generated tautomers (from 34 for compounds **2b** and **2c** to 86 for compounds **6a-6c**), were optimized quantum-mechanically. The optimized structures of the tautomers were subjected to conformational search with TorsiFlex. Because the conformational search led to a significant decrease of energy of certain tautomers (>20 kcal/mol for example of compound **8a**), it proved to be necessary to include in the conformational search all of the generated tautomers, instead of selecting some of them (e.g., top-10, or within 10 kcal/mol from the top tautomer).

The most stable conformation for compound **8a** found by TorsiFlex was of the ***E*-AAK** isomer. Chemically, *E*/*Z* isomerization of the double bond transforms it into a ***Z*-AAK** isomer. However, in terms of the conformational search, it is rotation around bond C21-C22 (according to Figure 1 labeling), that transforms the ***E*-AAK** isomer into ***Z*-AAK**. Multiple increases in the number of cycles of the conformational search led to the same ***E*-AAK** leading conformation, while the ***Z*-AAK** conformation was not found among the solutions. However, the presence in the results of other conformations requiring rotation around this bond but much higher in energy, proves that this torsion was included in the conformational search.

A manual flip around bond C21-C22 of the best conformational search solution transforming ***E*-AAK** into ***Z*-AAK,** followed by QM energy minimization, led to a 2.12 kcal/mol energy decrease, which is in agreement with the crystallographic results. Thus, formal generation of all possible tautomers followed by a conformational search required considerable computational resources but did not lead to the most stable configuration, while manual generation of possible isomers, tautomers, and major conformers proved to be more efficient.

The ***Z*-AAK** tautomer can be also transformed into ***Z*-HIK** via short-distance proton transfer. The energy profile of this process in vacuo was calculated for compound **8a** (Figure 2), yielding a 4.5 kcal/mol energy barrier and a 1.05 kcal/mol energy loss. Thus, the system predominantly exists in the ***Z*-AAK** form.

### 3.2. Biological Studies

#### 3.2.1. Esterase Profile Assessment

The method for evaluation of the esterase profile developed by our group includes the determination of the inhibitory activity of the synthesized compounds against enzymes of the cholinesterase family—AChE and BChE—as well as a structurally close enzyme–carboxylesterase (CES). Inhibition of AChE and BChE in the brain increases acetylcholine levels and improves cognitive functions in AD. CES is responsible for hydrolysis of numerous therapeutically important ester-containing drugs, and therefore inhibition of CES by anticholinesterase compounds used in AD therapy could lead to adverse drug-drug interactions.

Human erythrocyte AChE, equine serum BChE, and porcine liver CES were used to assess the esterase profile. The applicability of this set of enzymes for this purpose has been shown earlier [112,113,114,115]. The results on the esterase profile of model compounds **2** and conjugates **6–8** are displayed in Table 1.


The results presented in Table 1 indicate that compounds **2a-c** – *N*-hexylamine derivatives of 2-tolylhydrazinylidene-1,3-diketones **1a-c** inhibit cholinesterases very weakly and do not inhibit CES. At the same time, all 3 groups of conjugates of 2-tolylhydrazinylidene-1,3-diketones **1a-c** with tacrine **6**, **7**, and **8** exhibit high inhibitory activity against cholinesterases – at the level and above the parent pharmacophore tacrine with predominant inhibition of BChE and weak inhibition of CES.

Prior to the research, we feared that the synthesized conjugates **6, 7,** and **8** might exhibit anti-CES activity undesirable for AD therapy agents, because we have previously found that one of the starting compounds, 2-tolylhydrazinylidene-1,3-diketone **1c**, exhibits moderate inhibitory activity against this enzyme [79]. However, it turned out that the isomerization of the hydrazone tautomer to the azo form upon replacement of one of the carbonyl groups by the alkylamine functional group led to a significant decrease in the anti-CES activity.

*AChE inhibition***.** The structure of the R substituent at the carbonyl carbon atom (Me, Ph, or CF_3_) had practically no effect on the anti-AChE activity of conjugates. At the same time, for all series of conjugates **6**, **7**, and **8**, an increase in inhibitory activity against AChE was observed with an increase in the spacer length (from 4 to 8 CH_2_ groups). The anti-AChE activity of conjugates **6b,c**, **7b,c,** and **8b,c** was higher than the activity of the parent pharmacophore tacrine. Compounds **6c**, **7c**, **8b,** and **8c** exhibited the highest anti-AChE activity (IC_50_ = 0.27, 0.25, 0.28 and 0.24 µM, respectively).

*BChE inhibition***.** All conjugates were active against BChE. There was no pronounced effect of the spacer length on the inhibitory activity. However, anti-BChE activity depended on the structure of the R substituent at the carbonyl carbon atom: conjugates **7** with R = Ph were most active and reached the level of tacrine. The most effective BChE inhibitors were compounds **6c**, **7a**, **7b**, **7c** (IC_50_ = 0.054, 0.036, 0.047, and 0.0745 μM, respectively).

The selectivity toward BChE compared to AChE is maximum for compounds with a short spacer (CH_2_)_4_; it is especially pronounced for **7a** (R = Ph).

#### 3.2.2. Kinetic Studies of AChE and BChE Inhibition

The mechanism of inhibitory action of the conjugates toward AChE and BChE was studied using compound **6b** as a sample. The graphical analysis of the kinetic data on AChE (Figure 3A) and BChE (Figure 3B) inhibition by **6b** in the Lineweaver–Burk double-reciprocal plot demonstrates the changes in both *K*_m_ and *V*_max_ that attest to a mixed type of inhibition. The inhibition constants are as follows: *K*_i_ = 0.254 ± 0.018 µM (competitive component) and α*K*_i_ = 0.473 ± 0.041 µM (noncompetitive component) for AChE and *K*_i_ = 0.095 ± 0.008 µM (competitive component) and α*K*_i_ = 0.224±0.022 µM (noncompetitive component) for BChE.

#### 3.2.3. Molecular Modeling Studies

The most stable ***Z*-AAK** conformer of the considered compounds was taken for molecular docking studies. It was found that all compounds bind to the hAChE in a uniform way, occupying both the CAS and PAS. In the CAS, the protonated tacrine fragment binds forming π-π stacking interactions with the Trp86 side chain and a hydrogen bond with its main chain hydrogen oxygen (Figure 4A–C), as was observed previously for tacrine-containing conjugates [116,117]. Compounds **6a-c** also form a few hydrogen bonds between Tyr124 phenolic hydroxyl group and the linker nitrogen atoms or carbonyl atom of the acetyl group (Figure 4A). With increasing linker length, the tolyl substituent occupies more of the PAS, forming π-π or T-stacking interactions with Trp286. For compounds **7a-c** these interactions are with the phenyl group, which is advancing into the PAS and interacting with Trp286, instead of with the tolyl substituent, while the latter protrudes out of the gorge entrance. Additionally, the carbonyl oxygen atom of the keto-group forms hydrogen bonds with the Phe295 and Arg296 main chain nitrogen atoms (Figure 4B).

Regarding compounds **8a-c** with a trifluoromethylketone (TFK) group, specific interactions change with elongation of the linker. For compound **8a**, one of the fluorine atoms was found in the oxyanion hole. For compound **8b**, it forms hydrogen bonds with the Phe295 and Arg296 main chain nitrogen atoms. For compound **8c**, it interacts with Trp286 side chain, forming an F-π contact [118] (Figure 4C). Such modes of interactions were previously observed for a TFK compound, sliding down the gorge [119]. However, in the present case, no position with the carbonyl oxygen atom in the oxyanion site was found, which excludes the possibility of a subsequent covalent reaction between the enzyme and inhibitor.

For all the groups of inhibitors, increasing occupancy of the PAS with elongation of the linker can be associated with an increase in the propidium displacement ability.

In the case of BChE, binding is much more uniform with the carbonyl oxygen atom of the keto group in the oxyanion hole (Figure 5A–C). While ligands with increasing linker length occupy similar positions in the BChE active site, the tacrine fragment could form π-cation and π-π stacking interactions with Tyr332 or Trp82, supported by ionic interactions and hydrogen bonds with the Asp70 or His438 main chain oxygen atom, respectively. In the case of compounds **7a-c,** there are additional π-π or T-stacking interactions of the tolyl substituent with the Trp231 side chain (Figure 5B). In the case of compounds **8a-c**, the trifluoromethyl group forms F-π interactions with the Trp231 side chain (Figure 5C). Overall, the position of the TFK group of compounds **8a-c** in the catalytic site of BChE suggests the subsequent formation of a labile covalent bond and an adduct, called a tetrahedral intermediate analog [119].

#### 3.2.4. Displacement of Propidium from the *Ee*AChE PAS

The study of compounds as potential inhibitors of the proaggregant activity of AChE was carried out by assessment of the degree of displacement of the selective PAS ligand propidium iodide from the *Ee*AChE PAS. As mentioned above, AChE PAS interacts with soluble Aβ peptides, promoting their aggregation. In this regard, compounds blocking the AChE PAS are potential antiaggregant agents.

As seen from Table 1, conjugates of tacrine with 2-tolylhydrazinylidene-1,3-diketones at a concentration of 20 μM reduce the fluorescence intensity of the propidium iodide bound to *Ee*AChE and displace propidium at or above the level of the control compound donepezil (11–19%). These data indicate the potential ability of conjugates to block AChE-induced aggregation of β-amyloid, which agrees with the mixed type inhibition of AChE by the conjugates (Figure 3A) and the results of molecular docking (Figure 4A–C). On the whole, the ability to displace propidium in all groups of conjugates **6**, **7**, and **8** increases with spacer elongation in agreement with the molecular docking results. The conjugates **6a-c** (R = Me) exhibit the highest activity, displacing propidium from *Ee*AChE PAS in the range of 15.6–19%, while the optimal spacer length is n = 6.

#### 3.2.5. Antioxidant Activity

We have earlier shown [79] that 2-arylhydrazylidene-1,3-diketones **1c**,**d** exhibit high radical-scavenging activity in the ABTS assay. Herein, we studied the primary antioxidant activity of a series of model compounds – aminoenketones **2a-c**, and conjugates **6a-c**, **7a-c**, **8a-c**, obtained from diketone **1c** and its non-fluorinated analogs **1a,b**.

The antioxidant activity of the compounds **2a-c**, **6a-c, 7a-c, 8a-c** was assessed by spectrophotometric ABTS and FRAP tests. The ABTS assay evaluates the binding of a model ABTS radical cation (ABTS**^•^****^+^**), which is realized by the mechanism of single electron transfer (SET) and/or hydrogen atom transfer (HAT). The FRAP assay (Ferric Reducing Antioxidant Power) measures the ability of compounds to reduce the ferric 2,4,6-tripyridyl-s-triazine complex [Fe(TPTZ)_2_]^3+^ to [Fe(TPTZ)_2_]^2+^, which occurs exclusively by the SET mechanism. The results are presented in Table 2.

It was found that model compounds aminoenketones **2a-c,** which are the reaction products of the interaction of 2-arylhydrazinylidene-1,3-diketones **1a-c** (Figure 1) with hexylamine, exhibit high radical-scavenging activity in the ABTS assay at the level of the standard antioxidant Trolox. The activity of the compounds did not depend on the substituent R at the carbonyl carbon atom (CH_3_, Ph or CF_3_). Conjugates **6a-c, 7a-c, 8a-c,** which are a combination of 2-arylhydrazinylidene-1,3-diketones **1a-c** and the anticholinesterase pharmacophore tacrine retained high ABTS**^•+^**-scavenging activity of diketones at the Trolox level. The variation of the substituent R at the carbonyl carbon atom, as well as the spacer length, practically did not affect the radical-scavenging activity of the conjugates.

In the FRAP assay, the model compounds **2a,b** and conjugates **6a-c, 7a-c,** also demonstrated fairly good activity. However, their activity in the FRAP test was twice lower than in the ABTS test.

The most notable finding was that there was no influence of the structure of the R group in the carbonyl fragment on the activity in the ABTS test: both aminoenketone **2** and conjugates of 1,3-diketones with tacrine **6–8** were highly active. At the same time, in the FRAP test, compound **2c** with R=CF_3_ was not active and a weak ferric reducing ability was observed for conjugates **8a-c** with the CF_3_ substituent in the carbonyl moiety. Taking into account that the FRAP assay measures the Fe^3+^ reducing ability occurring exclusively by the SET mechanism, whereas the binding of the ABTS radical cation can be realized by SET and/or HAT mechanisms, the observed different structure-activity relationships (in particular, no decrease in activity in the ABTS test for compounds **8a****-****c** with a CF_3_ substituent) may indicate different mechanisms of the antioxidant action of the studied compounds in the ABTS and FRAP tests. That is, the SET mechanism may be involved in the ferric reducing test and the HAT may be operating in the scavenging of the ABTS radical cation.

#### 3.2.6. Metal-chelating Properties

It is known that decreasing excess concentrations of metals in the brain by chelating agents is one of the approaches to AD treatment [51,52]. The complexation abilities of compounds **6a, 7a, 8a** for biometals such as Cu^2+^, Fe^2+^ and Zn^2+^ in acetonitrile were studied by UV–Vis spectrometry [81] and the results for conjugates **6a, 7a, 8a** are shown in Figure 6A–C. In the UV-Vis spectra of **6a, 7a, 8a,** very broad bands from 340 to 470 nm and from 280 to 340 nm are ascribed to π-π* transitions, and the peak around 240 nm is due to intramolecular charge transfer.

The spectra of the individual ions and compounds were subtracted from the spectra of the mixtures of each compound with each ion. A red shift of absorbance calculated in this way indicates the formation of a ligand-ion complex in the mixture. For example (Figure 7), it can be seen that there are two red shifts of absorbance from 240 to 253 nm and from 328 to 339 nm for the interaction of conjugate **8a** with Zn^2+^. For additional graphics, please see Appendix A. The results obtained demonstrate metal-chelating ability for conjugates **6a, 7a**, and **8a** to all three biometal ions (Cu^2+^, Fe^2+^ and Zn^2+^).

#### 3.2.7. Prediction of ADMET and Physicochemical Profiles

The results of our computational estimates of selected ADMET and physicochemical properties for compounds **2, 6–8** are shown in Table 3. All of the compounds had high predicted values for intestinal absorption, enabling their oral administration. Moreover, we could expect reasonable CNS activity in view of rather high predicted blood–brain barrier permeability (brain concentration exceeds the plasma concentration), although some optimization of this parameter might be desirable. The cardiac toxicity risk parameters (hERG p*K_i_* and pIC_50_) fell within 4.2–7.6 log units for all the analyzed compounds, which was within the lower or medium part of their possible range (3–9 log units). According to the commonly accepted drug-likeness guidelines, the predicted lipophilicities and aqueous solubilities, as well as the molecular weights of the compounds, were within or close to the desirable range for potential drug compounds, although the LogP values in some cases violated the original Rule-of-5 limits (however, given that some of the compounds were outside of the model applicability domain, the predicted values were not fully reliable). The integral quantitative estimates of drug-likeness (QED) were in the 0.1–0.4 range.

Consequently, the predicted ADMET and physicochemical properties of the compounds were acceptable for potential lead compounds in the discovery phase. Nevertheless, additional studies and structure optimization would be desirable to help maximize safety and improve the pharmacokinetic profile.

## 4. Conclusions

In summary, we synthesized the new conjugates of tacrine modified with 2-tolylazo-1,3-aminoenketone moieties with various acyl substituents through alkylene spacers of different lengths. According to NMR spectroscopy, the compounds were characterized by a predominant existence in solutions as a (*Z*)-2-azo-1,3-aminoenketone tautomer. According to XRD, compound **8a** in crystals also exists in (*Z*)-2-azo-1,3-aminoenketone form that agrees with QM calculations.

In general, conjugates **6a-c, 7a-c, 8a-c** demonstrated high inhibitory activity against both cholinesterases AChE and BChE, with selectivity for BChE, being mixed type inhibitors of both cholinesterases, and with weak inhibition of CES.

The structure of the R substituent at the carbonyl carbon atom (Me, Ph, or CF_3_) had practically no effect on the anti-AChE activity of the conjugates. However, an increase in inhibitory activity was observed for all conjugates upon elongation of the alkylene spacer, in agreement with the results of molecular docking. Thus, a considerable group of compounds **6b,c, 7b,c, 8b,c** showed high anti-AChE activity exceeding the activity of tacrine: IC_50_ = 0.24-0.34 µM.

The synthesized conjugates demonstrated high inhibitory activity against BChE. Their binding in the BChE active site was more uniform than seen with AChE, with the carbonyl oxygen atom of the keto group in the oxyanion hole. The most active inhibitors of BChE were compounds **6c** and **7a-c** (IC_50_ = 0.036–0.0745 µM), which reached the activity of tacrine.

All conjugates **6–8** showed a good ability to displace propidium from *Ee*AChE PAS that along with the mixed type of inhibition revealed by the kinetics study and the molecular docking results indicates their potential ability to block AChE-induced aggregation of Aβ.

The synthesized conjugates have good antioxidant potency. All compounds **6–8** demonstrated the ability to effectively scavenge ABTS radical cation at the Trolox level; compounds **6, 7** also showed rather high Fe^3+^ reducing activity in the FRAP test. Moreover, metal-chelating ability for biometals such as Cu^2+^, Fe^2+^ and Zn^2+^ was demonstrated for conjugates **6a, 7a**, and **8a**.

Finally, computational predictions of physico-chemical and ADMET properties of the compounds were consistent with drug-like characteristics and low toxicity.

The obtained results allowed us to consider the synthesized conjugates as new multifunctional agents for the potential treatment of AD. In particular, the non-fluorinated conjugates **6b,c**, **7b,c** with (CH_2_)_6_ and (CH_2_)_8_ linkers exhibited the most promising set of properties and are recommended for further in-depth studies, e.g., lipid peroxidation in brain homogenates, neuroprotective activity in cell cultures, cognition enhancing efficacy in animal models, and safety and bioavailability assessments.

## Data Availability

Not applicable.

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
