# Peer review of "New Multifunctional Agents for Potential Alzheimer’s Disease Treatment Based on Tacrine Conjugates with 2-Arylhydrazinylidene-1,3-Diketones"

_biomolecules, 2022, doi:10.3390/biom12111551_

Round 1

Reviewer 1 Report

The overall study is unique and interesting. 

Please consider my following comments. 

Abstract:

Line 30: I suggest to replace the word "problem" with task as it is urgent and challenging task not the problem.

Introduction: 

Line 52: The first sentence is taken from just reference 1 (2020 AD facts and figures). What is the contribution of reference 2 and 3 here? More over the 1st paragraph has no citations except 1st line. Kindly add suitable references from where these details are extracted. 

Paragraph 2: There is redundancy of AD word. Please fix it.  

Line 58,59 and 60: Please rewrite this sentence to be more clear and specific. 

line 61: change "characteristics" to "characteristic" as you mentioned just one issue. 

Line 70: Isn't it more appropriate to replace "synapsis space" to " synaptic space" ?

Line 107-108: Please rewrite this sentence as it looks incomplete. " One of the plausible mechanisms contributing to AD pathogenesis and imbalance of the homeostasis of brain biometals such as Cu2+, Zn2+, and Fe2+." 

Line 112: Modify "and an increase of oxidative stress" to "and increase oxidative stress" 

line 113: I suggest to replace" one of the reasons" to " one of the reason. 

Line 154: All solvents, chemicals, and reagents were obtained commercially and used without purification. Please provide the details of suppliers in bracket as mentioned for instruments. 

Methodology, Results and conclusion is well written. 

Line 168: Modify "were synthesized by previously published methods" to "were synthesized by referring previously published methods"

 Suggestion: If possible, after the conclusion section, kindly mention the recommendations, to extend this work to confirm these findings. 

Author Response

Reviewer 1

Open Review

English language and style

( ) Extensive editing of English language and style required
( ) Moderate English changes required
(x) English language and style are fine/minor spell check required
( ) I don't feel qualified to judge about the English language and style

Yes

Can be improved

Must be improved

Not applicable

Does the introduction provide sufficient background and include all relevant references?

(x)

( )

( )

( )

Are all the cited references relevant to the research?

(x)

( )

( )

( )

Is the research design appropriate?

(x)

( )

( )

( )

Are the methods adequately described?

(x)

( )

( )

( )

Are the results clearly presented?

(x)

( )

( )

( )

Are the conclusions supported by the results?

(x)

( )

( )

( )

Comments and Suggestions for Authors

The overall study is unique and interesting. 

Please consider my following comments. 

Abstract:

Line 30: I suggest to replace the word "problem" with task as it is urgent and challenging task not the problem.

 Done.

Introduction: 

Line 52: The first sentence is taken from just reference 1 (2020 AD facts and figures). What is the contribution of reference 2 and 3 here? More over the 1st paragraph has no citations except 1st line. Kindly add suitable references from where these details are extracted. 

Refs [1,2] have been moved after sentence 3, Ref [3] has been changed.

Paragraph 2: There is redundancy of AD word. Please fix it.  

Done.

Line 58,59 and 60: Please rewrite this sentence to be more clear and specific. 

The sentence has been rewritten.

  • line 61: change "characteristics" to "characteristic" as you mentioned just one issue. 

 "Characteristics" is actually correct – it is the object of a preposition, and "cholinergic deficiency" is the selected attribute referred to by "One of ….".

Line 70: Isn't it more appropriate to replace "synapsis space" to " synaptic space" ?

Done.

Line 107-108: Please rewrite this sentence as it looks incomplete. " One of the plausible mechanisms contributing to AD pathogenesis and is imbalance of the homeostasis of brain biometals such as Cu2+, Zn2+, and Fe2+." 

Corrected.

Line 112: Modify "and an increase of oxidative stress" to "and increase oxidative stress"

Done.

line 113: I suggest to replace" one of the reasons" to " one of the reason”. 

"…one of the reasons…."  Is correct.

Line 154: All solvents, chemicals, and reagents were obtained commercially and used without purification. Please provide the details of suppliers in bracket as mentioned for instruments. 

Done.

Methodology, Results and conclusion is well written. 

Line 168: Modify "were synthesized by previously published methods" to "were synthesized by referring previously published methods"

Done.

 Suggestion: If possible, after the conclusion section, kindly mention the recommendations, to extend this work to confirm these findings.

Done.

Reviewer 2 Report

The work describes the design, synthesis, characterisation and evaluation of new multifunctional compounds with potential application for the treatment of Alzheimer’s disease (AD). The authors tried to exhaustively demonstrate the various potential applications of the synthesized molecules. I wonder why the authors did not try to publish two or three parts. The results were well described apart from some which I am mentioning below. I would recommend the publication of the work with minor revisions.

  1. The introduction is lengthy and they may need to merge some paragraphs.

  2. They have mentioned the use of quantum chemical calculations like HF and B3LYP in the methods section but they did not show the results neither text nor the supplementary section.

  3. They also mentioned the use of FT IR for the characterisation of the molecules but lack showing the results at least in the supplementary section. Since the molecules are new some readers may be interested.

  4. Some sections the topics need to be formatted  should be consistent.

Author Response

Reviewer 2

Open Review

English language and style

( ) Extensive editing of English language and style required
( ) Moderate English changes required
(x) English language and style are fine/minor spell check required
( ) I don't feel qualified to judge about the English language and style

Yes

Can be improved

Must be improved

Not applicable

Does the introduction provide sufficient background and include all relevant references?

(x)

( )

( )

( )

Are all the cited references relevant to the research?

(x)

( )

( )

( )

Is the research design appropriate?

(x)

( )

( )

( )

Are the methods adequately described?

(x)

( )

( )

( )

Are the results clearly presented?

(x)

( )

( )

( )

Are the conclusions supported by the results?

(x)

( )

( )

( )

Comments and Suggestions for Authors

The work describes the design, synthesis, characterisation and evaluation of new multifunctional compounds with potential application for the treatment of Alzheimer’s disease (AD). The authors tried to exhaustively demonstrate the various potential applications of the synthesized molecules. I wonder why the authors did not try to publish two or three parts. The results were well described apart from some which I am mentioning below. I would recommend the publication of the work with minor revisions.

  1. The introduction is lengthy and they may need to merge some paragraphs.

                        Done.

  1. They have mentioned the use of quantum chemical calculations like HF and B3LYP in the methods section but they did not show the results neither text nor the supplementary section.

Quantum chemical calculations were used for conformational search by the TorsiFlex program, calculations of tautomerization pathways energies (results reported in the very end of section 3.1), and geometry optimization of ligands prior to molecular docking (results reported in section 3.2.3).

  1. They also mentioned the use of FT IR for the characterisation of the molecules but lack showing the results at least in the supplementary section. Since the molecules are new some readers may be interested.

Done.

  1. Some sections the topics need to be formatted  should be consistent.

            In 2.2. Biological testing

            2.2.1. Enzymatic assays

            Section numbers 2.2.1.1 and 2.2.1.2 have been added

Reviewer 3 Report

The manuscript includes the synthesis of new compounds derived from  tacrine. Based on the ADMET and physicochemical profiles of synthesized compounds some of the new compounds are predicted  to exhibit higher blood–brain barrier permeability than tacrine.

Although the studies included in the present manuscript predict  a therapeutic potential of  the new synthesized molecules for AD management, many in vitro and in vivo studies need to be performed. The authors could include one or two sentences in conclusions section about some of those studies to be carried out. 

Inclusion of a graphical abstract would be welcome.

Author Response

Reviewer 3

Open Review

English language and style

( ) Extensive editing of English language and style required
( ) Moderate English changes required
(x) English language and style are fine/minor spell check required
( ) I don't feel qualified to judge about the English language and style

Yes

Can be improved

Must be improved

Not applicable

Does the introduction provide sufficient background and include all relevant references?

(x)

( )

( )

( )

Are all the cited references relevant to the research?

(x)

( )

( )

( )

Is the research design appropriate?

(x)

( )

( )

( )

Are the methods adequately described?

(x)

( )

( )

( )

Are the results clearly presented?

(x)

( )

( )

( )

Are the conclusions supported by the results?

(x)

( )

( )

( )

Comments and Suggestions for Authors

The manuscript includes the synthesis of new compounds derived from  tacrine. Based on the ADMET and physicochemical profiles of synthesized compounds some of the new compounds are predicted  to exhibit higher blood–brain barrier permeability than tacrine.

Although the studies included in the present manuscript predict  a therapeutic potential of  the new synthesized molecules for AD management, many in vitro and in vivo studies need to be performed. The authors could include one or two sentences in conclusions section about some of those studies to be carried out. 

-- Done.

  • Inclusion of a graphical abstract would be welcome.

-- A graphical abstract will be uploaded with the revised manuscript.

-- Abbreviated names of all tautomers are given in font BOLD. The changes from normal to bold are marked with track changes.